# Ultra-Short-Term and Short-Term Heart Rate Variability Recording during Training Camps and an International Tournament in U-20 National Futsal Players

**DOI:** 10.3390/ijerph17030775

**Published:** 2020-01-26

**Authors:** Yung-Sheng Chen, Filipe Manuel Clemente, Pedro Bezerra, Yu-Xian Lu

**Affiliations:** 1Department of Exercise and Health Sciences, University of Taipei, Taipei 111, Taiwan; yschen@utaipei.edu.tw (Y.-S.C.); whitelu0422@gmail.com (Y.-X.L.); 2Escola Superior Desporto e Lazer, Instituto Politécnico de Viana do Castelo, Rua Escola Industrial e Comercial de Nun’Álvares, 4900-347 Viana do Castelo, Portugal; pbezerra@esdl.ipvc.pt; 3Instituto de Telecomunicações, Delegação da Covilhã, 1049-001 Lisboa, Portugal; 4The Research Centre in Sports Sciences, Health Sciences and Human Development, 5001-801 Vila Real, Portugal; 5Graduate Institute of Athletes and Coaching Science, National Taiwan Sport University, Taoyuan 33301, Taiwan

**Keywords:** futsal, performance, heart rate variability, training load, general wellness, autonomic function

## Abstract

The aim of this study was to examine ultra-short-term and short-term heart rate variability (HRV) in under-20 (U-20) national futsal players during pre-tournament training camps and an official tournament. Fourteen male U-20 national futsal players (age = 18.07 ± 0.73 yrs; height = 169.57 ± 8.40 cm; body weight = 64.51 ± 12.19 kg; body fat = 12.42% ± 3.18%) were recruited to participate in this study. Early morning 10 min resting HRV, Borg CR-10 scale session rating of perceived exertion (sRPE), and general wellness questionnaire were used to evaluate autonomic function, training load, and recovery status, respectively. Log-transformed root mean square of successive normal-to-normal interval differences (LnRMSSD) was used to compare the first 30 s, first 1 min, first 2 min, first 3 min, and first 4 min with standard 5 min LnRMSSD. Mean (LnRMSSD_mean_) and coefficient of variation (LnRMSSD_cv_) of LnRMSSD were used to compare the different time segments of HRV analysis. The result of LnRMSSD_mean_ showed nearly perfect reliability and relatively small bias in all comparisons. In contrast, LnRMSSD_cv_ showed nearly perfect reliability and relatively small bias from 2-4 min time segments in all study periods. In conclusion, for accuracy of HRV measures, 30 s or 1 min ultra-short-term record of LnRMSSD_mean_ and short-term record of LnRMSSD_cv_ of at least 2 min during the training camps are recommended in U-20 national futsal players.

## 1. Introduction

Futsal is a high-intensity and competitive indoor sport [1]. This sport requires energy cost in mixed aerobic and anaerobic consumption, strenuous physical contacts, locomotion activities, and skill acquisitions [2]. The intensive nature of the futsal competition is extremely high and very often causes physiological and psychological stress to the players [3]. It has been reported that high-speed running (18.1–25 km^.^h^-1^) and sprint (> 25 km^.^h^-1^) is about 22.6% of the total covering distance. Also, the average heart rate (HR) responses reach to 170-190 beats^.^min^–1^ in Liga Nacional de Futbol Sala (around 83% of maximal HR during the match time) [1]. Locomotion activities of the futsal match change in every 8–9 s, indicating intermittent patterns of physical activity [4].

Heart beat (HR) rhythm is regulated by the autonomic nervous system (ANS). Assessing the R-R intervals (RRI) variation of HR beats in a set of time series can be used to understand HR variability (HRV), which indicates the strength of sympathetic and parasympathetic activities [5,6]. The characteristics of HRV modulation after exercise depend upon intensity, duration, and modality of exercises [7,8,9]. The variation of HRV indices is also associated with training adaptation to aerobic capacity after longitudinal endurance training [10] and preseason futsal training [11].

Ultra-short-term HRV refers to an extremely short time segment of RRI record from consecutive heart rate beats less than 1 min, while short-term time segment of RRI record refers to an HRV measure lasting 2-5 min duration. An ultra-short-term HRV record provides time efficiency of data collection in sports training [12,13], psychological measure [14,15], and clinical proposal [16]. Previous studies suggested that a 1 min ultra-short-term HRV record can be used as a surrogate to a traditional 5 min short-term HRV record [12,13,17,18,19,20]. Esco and his colleagues [17] reported that the ultra-short-term record of natural algorithm of root mean square of successive RRI (LnRMSSD) is suggested to assess 1 min time segment in order to avoid errors of measures in athletic population. Moreover, excellent limit of agreement and accuracy of 1 min ultra-short-term record of LnRMSSD measures was observed during a 5 min stabilization period in athletes [13,21].

Monitoring quantity and intensity of training workloads becomes a primary consideration in periodization and injury prevention of sports participation [22]. However, evaluation of recovery status is also receiving great attention in strength and conditioning coaches and sports scientists to optimize players’ performance and understand the most appropriate periods to apply the right stimulus [23,24]. Session rating of perceive exertion (sRPE) and general wellness questionnaire (e.g., Hooper) are two common tools to assess the psychophysiological stress in responses to the training workload and individual perception of recovery status [25]. Measuring daily change of resting HRV can be used to understand the autonomic modulation in relation to consequence of training adaptation [10,26]. The modulation of HRV during training period is in adjunction with intensity of training load (TL) and recovery status in daily football or futsal training sessions [27,28,29]. 

The standard process to record short-term HRV suggests measuring at least 5 min, followed by a 5 min stabilization [30]. Recent studies suggest that ultra-short-term HRV assessment can provide valid and reliable information for monitoring autonomic adaptation to sports training while LnRMSSD is used as the parameter [12,13,18,20]. Studies conducted by Nakamura et al. [20] and Pereira et al. [13] investigated the ultra-short-term HRV of elite futsal players who undertook futsal training and matches in seasonal periodization. In contrast, national futsal teams organized the tournament preparation based on irregular short-term training camps due to limitations of player recruitment, budget, and team schedule. Currently, there are no studies to investigate the ultra-short-term HRV during tournament preparation and tournament competition in under-20 (U-20) futsal players. Therefore, the first aim of this study was to compare the degree of validity and agreement of ultra-short-term and short-term HRV records of LnRMSSD variable (0-4 min) with standard 5 min LnRMSSD record during three pre-tournament training camps and an official tournament in U-20 national futsal players. The second aim of this study was to determine the relationship between the training load (TL) and general wellness during these four periods. On the basis of previous studies, we hypothesized that validity and agreement of ultra-short-term and short-term LnRMSSDs decrease during the study periods with high TL and low wellness scores.

## 2. Materials and Methods

### 2.1. Subjects

Fourteen male Chinese Taipei U-20 national team futsal players were recruited in this study. Twelve field players and two goalkeepers were included in this study (mean ± standard deviation: age = 18.07 ± 0.73 yrs; height = 169.57 ± 8.40 cm; body weight = 64.51 ± 12.19 kg; body fat = 12.42% ± 3.18%). The data were collected during the first training camp (TC_1st_; *n* = 13; July 10^th^ and 15^th^ 2018), invitation tournament (IT; *n* = 11; July 28^th^ and August 2^nd^; 2018); second training camp (TC_2nd_; *n* = 14; November 6^th^ and 11^th^ 2018), and official tournament of 2018 AFC U-20 Futsal Championship East Asia Qualified Games (OT; *n* = 14; November 27^th^ and December 2^nd^ 2018). One goalkeeper did not attend the TC_1st_ and IT, while two field players did not attend the IT period. All participants signed an informed consent form and were familiarized with experimental procedures. The study has been approved by the human ethics committee of University of Taipei and was conducted in according to the Declaration of Helsinki.

### 2.2. Experimental Procedure

The team schedule consisted of two domestic training camps (TC_1st_ = 7 days, 8 training sessions; TC_2nd_ = 7 days, 11 training sessions), one oversea invitation tournament (IT = 6 days, 1 training session, 4 matches), and one official tournament (OT = 8 days, 3 training sessions, 3 matches) (Table 1). The players’ morning resting HRV, general wellness questionnaire, and sRPE were measured during these four periods. The players’ height and weight were measured via a portable stadiometer (Seca 213, SECA, Germany) and electrical weight scale (Xyfwt382, Teco, Taiwan) in the registration day of training camp. Four skinfold thickness measurements were used to assess the percentage of body fat via a skin folder (Lange Skinfolder Caliper, Beta Technology, USA). The percentage of body fat was obtained by using the formula 5.783 + 0.153 * (the sum of triceps, subscapular, suprailiac, abdominal skinfolds) / 100 [31]. For the HRV data collection, a portable Polar HR monitor (Polar team Pro, Polar Electro, Kemple, Finland) was mounted onto the participant’s front chest to record resting HRV in a sitting position. The participants were instructed to control breath with self-controlled patterns. After 10 min resting HRV record, the players reported the score of general wellness questionnaire. The measurements were taken in a quiet and spacious room between 7 a.m. and 8 a.m. For TL monitoring, sRPE was used to record TL during the training sessions and matches. The participants reported the individual perception of TL to the sport trainer within 30 min after completion of the training sessions. During the invitation and official matches, individual sRPE was reported to the sport trainer in the dressing room after the end of matches. A qualified sports trainer conducted the anthropometric measurement and collected all data during these periods. The methods to record anthropometric measurement, resting HRV, sRPE, and general wellness questionnaire have been reported in our previous study [32].

### 2.3. Heart Rate Variability

All participants were required to measure the resting HRV in a sitting position in the morning prior to the breakfast. Participants sat on chairs in a comfortable position for 5 min, followed by 5 min data collection. A telemetric HR monitor system was used to record the resting HRV (Polar team Pro, Polar Electro, Kemple, Finland). The HRV data were exported to Polar team pro web service and then extracted to personal laptop for data analysis. Kubios HRV analysis software (Premium version 3.2, Kubios, Kuopio, Finland) was used to calculate LnRMSSD. Medium artefact correction and smoothing priors set at 500 Lambda were used for HRV analysis [33]. The time segments of HRV records were divided into first 30 s (LnRMSSD_30s_), first 1 min (LnRMSSD_1min_), first 2 min (LnRMSSD_2min_), first 3 min (LnRMSSD_3min_), first 4 min (LnRMSSD_4min_), and standard 5 min LnRMSSD (LnRMSSD_5min_). Average (LnRMSSD_mean_) and coefficient of variation (LnRMSSD_cv_) of daily LnRMSSD during each training camp or tournament were used to compare the different time segment of HRV analysis.

### 2.4. Training Load

The perceived exertion of players after the training session was assessed by using the Borg CR-10 scale [34]. The players were asked to answer the question “how intense was the session?” using a visual analog scale in which 0 means “nothing at all” and 10 means “extremely strong”. After collection of the player’s answers, the score was multiplied by the time of the training session in minutes, thus providing the sRPE in arbitrary units (a.u.) [35]. An educational session to familiarize the use of sRPE was organized for all participants in the registration day of the training camp. The answers were provided individually to avoid the players hearing their colleagues’ scores.

### 2.5. Recovery Status

The general wellness questionnaire [36] was used to assess recovery status. The general wellness questionnaire consists of five components to assess fatigue, sleep quality, muscle soreness, stress, and mood status. Each component consists of five points, in which the highest score (i.e., point 5) represents the better state and the lowest score (i.e., point 1) represents the worst state. The sum of five components scored (lowest score: 5 points, highest score: 25 points) was calculated to evaluate the general aspect of fatigue and recovery status [36]. The answers were provided individually to avoid the players hearing their colleagues’ scores.

### 2.6. Statistical Analyses

Statistical analyses were conducted using SPSS^®^ Statistics version 25.0 (IBM, Armonk, NY, USA) and Microsoft Excel 2013 (Microsoft Corporation, Redmond, WA, USA). Descriptive data of the measured variables are presented as mean ± standard deviation (SD) and 90% confidence intervals (90% CI). The intra-subject coefficient of variation of LnRMSSD during each study period was used to calculate the LnRMSSD_cv_ [37]. Between training camps/tournament differences of ultra-short-term and short-term LnRMSSD were analyzed by using the standardized differences of the effect size (ES). The level of ES was interpreted as trivial (0.0–0.2), small (0.2–0.6), moderate (0.6–1.2), large (1.2–2.0), very large (> 2.0) [37]. Interclass correlation coefficients (ICC) with two-way random model and single measure was used to determine relative values of reliability. The level of ICC values was expressed as nearly perfect (0.9–1), very large (0.70–0.89), large (0.50–0.69), moderate (0.31–0.49), and small (0–0.3) [38]. Probabilities were calculated via 0.2 * between-participants standard deviation (smallest worthwhile changes, SWC) [38]. Qualitative probabilistic mechanistic inferences about the true effects were made using the scale of qualitative probabilities as follows: 25–75% = possible; 75–95% = likely; 95–99% = very likely; > 99% = almost likely. Relationships between TL and general wellness were assessed by using Pearson’s product-moment correlation (*r*). The magnitude of the correlation coefficients was determined as trivial (*r* < 0.1), small (0.1 < *r* < 0.3), moderate (0.3 < *r* < 0.5), high (0.5 < *r* < 0.7), very high (0.7 < *r* < 0.9), nearly perfect (*r* > 0.9), and perfect (*r* = 1) [38]. Lastly, Bland–Altman plots were used to evaluate the upper and lower limits of agreements among time segments of LnRMSSD [39].

## 3. Results

### 3.1. Heart Rate Variability

The mean ± SD of LnRMSSD_mean_ and LnRMSSD_cv_ for all time segments comparisons during four study periods is presented in Figure 1. 

For LnRMSSD_mean_, it was found that the ES and ICC values showed trivial and nearly perfect in all comparisons, respectively. In addition, limits of agreements showed relatively small values of bias in all comparisons (Table 2).

For LnRMSSD_cv_, the result demonstrated a large variation of ES in all comparisons (from 0.00 - -0.91). The ICC values showed nearly perfect in LnRSSMD_cv2min_, LnRSSMD_cv3min_, and LnRSSMD_cv4min_ comparisons despite the type of camps (ICC range = 0.93 - 1). In addition, limits of agreements showed relatively small value of bias in LnRSSMD_cv4min_ across four study periods (Table 3). 

### 3.2. Session Rating of Perceived Exertion and General Wellness Score

The results showed that lowest daily TL was found in IT (334.90 ± 44.21 a.u.), whereas highest daily TL was found in TC_2nd_ (906.29 ± 93.60 a.u.). The daily TL in TC_1st_ and OT was 553.15 ± 114.84 (a.u.) and 482.75 ± 81.22 (a.u.), respectively. Qualitative probabilities for effect magnitude of TL were revealed as most likely large in TC_1st_ vs. IT (209.81; 23.05) and TC_2nd_ vs. OT (421.17; 1.9). In contrast, most likely small probabilities of mean difference were found in TC_1st_ vs. TC_2nd_ (-363.56; 8.30), IT vs. TC_2nd_ (-573.38; 1.20), and IT vs. OT (-152.20; 12.50) (Table 4).

The results of general wellness score showed low daily score in TC_1st_ (16.74 ± 1.61), TC_2nd_ (16.63 ± 2.44), and OT (17.18 ± 0.98). In contrast, high daily score was observed in IT (20.00 ± 3.18). The probabilities of mean difference were revealed as most likely large in IT vs. TC_2nd_ (3.69; 1.6) and IT vs. OT (2.81; 1.8). In contrast, most likely small probability of mean difference was identified in TC_1st_ vs. IT (-3.24; 0.5) (Table 4).

Results of Pearson’s correlation between TL and general wellness revealed that trivial positive correlation was found in TC_1st_ (*r* = 0.07; *p* = 0.81; CI 90%: −0.42 to 0.53). Small negative correlation was found in OT (*r* = −0.21; *p* = 0.48; CI 90%: −0.61 to 0.28). Moderate negative correlation was observed in IT (*r* = −0.53; *p* = 0.09; CI 90%: −0.82 to −0.01) and TC_2nd_ (*r* = −0.53; *p* = 0.05; CI 90%: −0.79 to -0.09) (see Figure 2).

## 4. Discussion

This study is the first to report the reliability and degree of agreement of ultra-short-term and short-term HRV during national team training camps and an official tournament in U-20 futsal players. The main findings in the present study were that ultra-short-term LnRMSSD_mean_ 30 s or 1 min provided valid and accurate to estimate the vagal-related change of autonomic activity despite TL and functions of training camps and tournaments. However, a shorter time segment of LnRMSSD_cv_ record less than 2 min showed large bias and invalid outcome, compared with standard 5 min measure. Moreover, correlation between TL and general wellness score was trivial in the first training camp and was a negatively small correlation during the official tournament. A moderate negative relationship between TL and general wellness score was found during the invitation tournament and training camp with high TL. The implementation of LnRMSSD parameter for ultra-short-term (within 1 min) and short-term (2-4 min) HRV records to monitor the training adaptation and recovery status during training camps and official tournament is warranted.

To date, this is first study to report ultra-short-term and short-term LnRMSSD in U-20 national futsal players during different TLs of training camps and a continental tournament. Our findings revealed that LnRMSSD_mean_ showed no large variability and disagreement of measures in all time segments across the study periods, as evidence by nearly perfect of ICC values (0.96 – 1.00) and relatively narrow values of limits of agreements (0.00 - 0.09) in all comparisons. It is well documented that LnRMSSD_mean_ can be used to indicate training adaptation of vagal-related activities in association with enhancement of aerobic capacity [10,11,40]. Accurate and reliable measure of 1 min ultra-short-term HRV as indicated by mean of LnRMSSD has been reported in rugby players [41], futsal players [13,20], youth female basketball players [12], collegiate cross-country athletes [21], and collegiate soccer and basketball players [17]. These studies reported excellent validity and sensitivity of 1 min ultra-short-term HRV record during a 5 min stabilization period. For nonathletic population, Krejčí et al. [42] demonstrated that 90 s of LnRMSSD_mean_ record can provide strong agreement of HRV measurements in collegiate students. It is interesting to note that these studies conducted the HRV measures in pre and post-training periods or a cross-sectional time point. In contrast, our study collected the HRV measures on a longitudinal basis (four study periods in 5 months). Collectively, our findings extended the notion in consistence of excellent validity and agreement of ultra-short-term LnRMSSD record during domestic training camps, oversea invitation tournament, and official tournament in U-20 male national futsal players when mean value of LnRMSSD was used. 

Another major finding in our study demonstrated that LnRMSSD_cv_ had large bias and could potentially cause invalid and inaccurate measure if the record duration was less than 2 min during the training camps (ICC: lowest 0.55, highest 0.97; Bias: lowest −3.04, highest −0.60). However, validity and reliability of ultra-short-term and short-term LnRMSSD_cv_ was observed during OT (ICC: 0.95–0.99; Bias: −0.08–−1.46). LnRMSSD_cv_ is a sensitive parameter that can be used to understand the daily variation of autonomic adaptations in response to TL and fatigue status [43]. The paradoxical relationship between LnRMSSD_mean_ and LnRMSSD_cv_ provides a true measurement of autonomic adaptation to sports training and avoids the misinterpretation of daily fluctuation of vagal-related changes and psychometric status [44]. Nakamura et al. [37] recently reported that increase in ultra-short-term records of LnRMSSD_mean_ and decrease in LnRSSMD_cv_ were associated with improving performance of Yo-Yo intermittent recovery level 1 test after 4 weeks preseason training in Portuguese professional futsal players. Nakamura et al. recorded the LnRMSSD 3 times per week, whilst our study recorded daily morning of resting LnRMSSD through the training camps. It is arguably that the decrease in number of measures may result in a loss of sensitivity to detect the variation of entire measure [45]. Thus, HRV should be recorded for at least 2 min to avoid the potential bias of individual variant of resting HRV when coefficient of variation of LnRMSSD is used to monitor the autonomic function during training camps. 

The TL and recovery status are primary factors to affect the resting HRV records in athletes. Our finding demonstrated that correlation analysis between TL and general wellness score showed trivial in the first training camp and was a negatively small correlation during the official tournament. A negative relationship between TL and general wellness score was found during the invitation tournament and training camp with high TL. It is accepted that higher TL and lower general wellness score is associated with lower cardiac-vagal tone and vice versa [45,46].

Our results showed low variation of LnRSSMD_cv_ (4.38% ± 2.20%–7.42% ± 4.05%) among all time segments during TC_2nd_, which displayed largest TL (906.29 ± 93.60 a.u.) among the study periods. This finding was against the previous studies that demonstrated large LnRSSMD_cv_ in association with high TL [11,40]. One possible explanation to this finding may be related to good physical preparation prior to the OT, as evidenced by highest LnRMSSD_mean_ (4.22 ± 0.32–4.26 ± 0.31 log) among the study periods. This finding was consistent with our recent study that reported that the LnRSSMD and general wellness score were not associated with accumulation of TL in adult national futsal players during a five-day oversea training camp prior to a continental tournament [32]. Nevertheless, our findings suggested that the bias and variation of LnRSSMD is not accompanied with TL and recovery status during training camps and tournament.

The limitations in this study were twofold. Firstly, the number of players for statistical analyses were unequal during the study periods. As there was a limited number of final registration players in the official competition, data from 14 players called up for final list of the tournament were used for data analysis. The players who were not selected in the final squad involved in domestic training camps and oversea invitation tournament were excluded. Interpretation of such information to team sports with a large number of squad (i.e., rugby or soccer) should be cautioned because of small samples in futsal teams. Secondly, we did not conduct fitness assessments across the study periods due to time constraints of the team schedule. We were unable to evaluate information regarding the initial status of physical capacities in relation to physiological and psychometric adaptations to the study periods. Thus, we limited the sample size and fitness assessment in this study. Implementation of submaximal intensity of fitness assessments on the arrival day of each training camp could be convenient as a friendly alternative to subsequent training sessions. Future studies should continue to analyze ultra-short-term and short-term HRV assessments in this population and identify the physical adaptations as a co-variable to explain some of the possible findings. Despite that, the study presents new and interesting findings in a growing team sport.

For practical implication, HRV measure has become a popular tool to evaluate the health and recovery status of autonomic function in athletes due to its positive correlation with fatigue, overtraining syndrome, and training adaptation in aerobic capacity. Time management is critical for elite sports teams. The routine of physiological and psychological evaluations in team sports requires time efficacy and convenience to athletes and coaches. Since the categories of under-age international competition have grown dramatically in recent years, implementation of LnRMSSD measure of at least 30 s can be considered to detect daily variation of cardiac autonomic functions as an easy and convenient alternative to practitioners and coaches in sports teams. 

## 5. Conclusions

In conclusion, for accuracy of HRV measures, 30 s and 1 min LnRMSSD_mean_ records after 5 min stabilization were revealed as a valid and reliable assessment for training adaptation of autonomic functions during short-term training camps and a tournament in U-20 national futsal players. The current finding suggested that measuring LnRMSSD_mean_ for 30 s or 1 min after postural stabilization during daily morning assessment was acceptable despite the types of training camp. Moreover, consideration should be taken when LnRMSSD_cv_ is used to evaluate the cardiac-autonomic activity in response to training loads. Short-term record of LnRMSSD_cv_ of at least 2 min during the training camps and an official tournament was suggested for use in young adult national futsal players.

## Figures and Tables

**Figure 1 ijerph-17-00775-f001:**
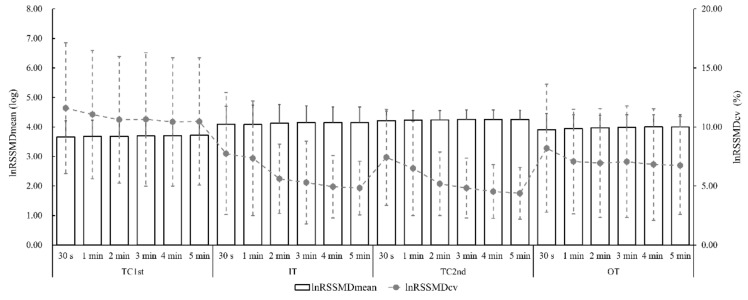
Mean and standard deviation of average (LnRMSSD_mean_) and coefficient of variation (LnRMSSD_cv_) of natural algorithm of root mean square of successive R-R intervals in first training camp (TC_1st_), invitation tournament (IT), second training camp (TC_2nd_), and official tournament (OT). Ultra-short-term (30 seconds and 1 minute) and short-term (2 minute, 3 minute, 4 minute, and 5 minute) records are presented in each study period.

**Figure 2 ijerph-17-00775-f002:**
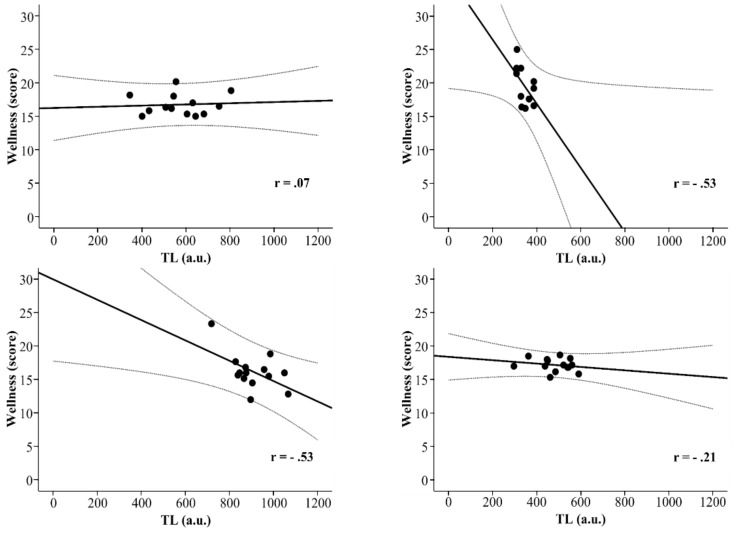
Pearson correlation between session rating of perceived exertion and general wellness score in first training camp (TC_1st_), invitation tournament (IT), second training camp (TC_2nd_), and official tournament (OT). sRPE = session rating of perceived exertion; TL = training load; AU (arbitrary units).

**Table 1 ijerph-17-00775-t001:** The schedule of domestic training camps, invitation tournament, and official tournament.

Periods	Day 1	Day 2	Day 3	Day 4	Day 5	Day 6	Day 7	Day 8
TC_1st_	Registration and TD (116 min)	TD (116 min)	TD (115 min)	TD (109 and 121 min)	TD (116 min)	TD (132 min)	TD (113 min)	
IT	Travelling to Shenzhen and TD (115 min)	FM vs. Hong Kong; 5:2; Win (89 min)	FM vs. Macau; 10:0; Win (100 min)	FM vs. Shenzhen; 5:0; Win (101 min)	FM vs. Hong Kong; 5:3; Win (91 min)	Travelling to Taiwan		
TC_2nd_	TD (136 min)	TD (115 min and 103 min)	TD (121 min and 121 min)	TD (99 min)	TD (121 min and 118 min)	TD (116 min and 100 min)	TD (121 min)	
OT	Travelling to Mongolia	TD (123 min)	TM vs. Japan; 10:1; Lost (101 min)	TD (94 min)	TD (48 min)	OM vs. Mongolia; 3:1; Win (86 min)	OM vs. China; 7:3; Win (97 min)	Travelling to Taiwan

TD = training day; FM = friendly match; TM = training match; OM = official match; TC_1st_ = first training camp; IT = invitation tournament; TC_2nd_ = second training camp; OT = official tournament.

**Table 2 ijerph-17-00775-t002:** Mean of natural logarithm of the root mean square differences between adjacent normal R-R intervals during 30 seconds, 1 minute, 2 minute, 3 minute, 4 minute, and 5 minute of time segments in first training camp, invitation tournament, second training camp, and official tournament.

Study Periods	Time Segments	ES (90% CI)	ICC (90% CI)	Bias ( ± 1.96_*_SD)
TC_1st_ (*n* = 13)	Standard 5 min	-	-	-
	0–30 s	0.11 (−0.53; 0.76)	0.98 (0.95; 0.99)	0.07 (−0.19; 0.32)
	0–1 min	0.07 (−0.57; 0.72)	0.99 (0.97; 1.00)	0.05 (−0.14; 0.24)
	0–2 min	0.08 (−0.57; 0.72)	0.99 (0.98; 1.00)	0.04 (−0.09; 0.18)
	0–3 min	0.06 (−0.59; 0.70)	1.00 (0.99; 1.00)	0.03 (−0.08; 0.14)
	0–4 min	0.02 (−0.63; −0.67)	1.00 (1.00; 1.00)	0.01 (−0.03; 0.06)
IT (*n* = 11)	Standard 5 min	−	−	−
	0–30 s	0.08 (−0.62; 0.78)	0.98 (0.96; 0.99)	0.05 (−0.23; 0.33)
	0–1 min	0.11 (−0.59; −0.82)	0.98 (0.95; 0.99)	0.06 (−0.24; 0.37)
	0–2 min	0.05 (−0.65;−0.75)	0.99 (0.96; 1.00)	0.03 (−0.25; 0.30)
	0–3 min	0.00 (−0.70; −0.70)	1.00 (0.99; 1.00)	0.01 (−0.11; 0.10)
	0–4 min	0.00 (−0.70; −0.70)	1.00 (1.00; 1.00)	0.00 (−0.05; 0.04)
TC_2nd_ (*n* = 14)	Standard 5 min	−	−	−
	0–30 s	0.12 (−0.50; 0.75)	0.98 (0.95; 0.99)	0.04 (−0.13; 0.20)
	0–1 min	0.09 (−0.53; −0.72)	0.99 (0.96; 0.99)	0.02 (−0.12; 0.17)
	0–2 min	0.06 (−0.56; −0.69)	1.00 (1.00; 1.00)	0.02 (−0.06; 0.09)
	0–3 min	0.00 (−0.62; −0.62)	1.00 (1.00; 1.00)	0.00 (−0.05; 0.05)
	0–4 min	0.00 (−0.62; −0.62)	1.00 (1.00; 1.00)	0.00 (−0.03; 0.03)
OT (*n* = 14)	Standard 5 min	−	−	−
	0–30 s	0.18 (−0.44; −0.81)	0.96 (0.88; 0.98)	0.09 (−0.28; 0.45)
	0–1 min	0.13 (−0.49; −0.76)	0.98 (0.95; 0.99)	0.05 (−0.14; 0.25)
	0–2 min	0.05 (−0.57; −0.67)	1.00 (0.99; 1.00)	0.02 (−0.07; 0.12)
	0–3 min	0.02 (−0.60; −0.65)	1.00 (1.00; 1.00)	0.01 (−0.06; 0.08)
	0–4 min	0.00 (−0.62; −0.62)	1.00 (1.00; 1.00)	0.00 (−0.05; 0.04)

s = seconds; min = minutes; SD = standard deviation; ES = effect size; CI = confident interval; ICC = intraclass correlation coefficient; TC_1st_ = first training camp; IT = invitation tournament; TC_2nd_ = second training camp; OT = official tournament.

**Table 3 ijerph-17-00775-t003:** Coefficient of variation of natural logarithm of the root mean square differences between adjacent normal R-R intervals during 30 seconds, 1 minute, 2 minute, 3 minute, 4 minute, and 5 minute of time segments in first training camp, invitation tournament, second training camp, and official tournament.

Study Periods	Time Segments	ES (90% CI)	ICC (90% CI)	Bias ( ± 1.96_*_SD)
TC_1st_ (*n* = 13)	Standard 5 min	-	-	-
	0–30 s	−0.20 (−0.85; 0.44)	0.83 (0.55; 0.94)	−1.13 (−9.24; 6.98)
	0–1 min	−0.10 (−0.76; 0.54)	0.97 (0.93; 0.99)	−0.60 (−3.97; 2.78)
	0–2 min	−0.02 (−0.67; 0.62)	0.99 (0.98; 1.00)	−0.16 (−2.05; 1.73)
	0–3 min	−0.03 (−0.68; 0.61)	0.99 (0.98; 1.00)	−0.19 (−2.29; 1.92)
	0–4 min	0.00 (−0.64; 0.65)	1.00 (0.99; 1.00)	0.02 (−1.39; 1.44)
IT (*n* = 11)	Standard 5 min	−	−	−
	0–30 s	−0.70 (−1.45; 0.00)	0.65 (0.06; 0.87)	−2.92 (−9.98; 4.15)
	0–1 min	−0.64 (−1.38; 0.07)	0.69 (0.15; 0.89)	−2.52 (−8.88; 3.84)
	0–2 min	−0.29 (−1.00; 0.41)	0.94 (0.77; 0.98)	−0.78 (−2.85; 1.29)
	0–3 min	−0.15 (−0.86; 0.55)	0.94 (0.84; 0.98)	−0.47 (−3.14; 2.21)
	0–4 min	−0.04 (−0.75; 0.66)	0.99 (0.97; 1.00)	−0.11 (−1.21; 0.98)
TC_2nd_ (*n* = 14)	Standard 5 min	−	−	−
	0–30 s	−0.91 (−1.58;−0.27)	0.55 (−0.07; 0.82)	−3.04 (−9.22; 3.13)
	0–1 min	−0.64 (−1.30;−0.01)	0.67 (0.15; 0.87)	−2.11 (−7.84; 3.62)
	0–2 min	−0.32 (−0.95;−0.30)	0.93 (0.74; 0.98)	−0.81 (−2.89; 1.28)
	0–3 min	−0.18 (−0.81;−0.44)	0.95 (0.87; 0.98)	−0.45 (−2.34; 1.44)
	0–4 min	−0.07 (−0.69;−0.56)	0.99 (0.98; 1.00)	−0.15 (−1.02; 0.72)
OT (*n* = 14)	Standard 5 min	−	−	−
	0–30 s	−0.29 (−0.93; 0.33)	0.95 (0.76; 0.98)	−1.46 (−4.61; 1.69)
	0–1 min	−0.08 (−0.70; 0.54)	0.98 (0.95; 0.99)	−0.33 (−2.78; 2.12)
	0–2 min	−0.06 (−0.68; 0.56)	0.99 (0.98; 1.00)	−0.20 (−1.57; 1.16)
	0–3 min	−0.07 (−0.69; 0.55)	0.99 (0.97; 1.00)	−0.31 (−2.08; 1.46)
	0–4 min	−0.02 (−0.64; 0.60)	0.99 (0.98; 1.00)	−0.08 (−1.60; 1.44)

s = seconds; min = minutes; SD = standard deviation; ES = effect size; CI = confident interval; ICC = intraclass correlation coefficient; TC_1st_ = first training camp; IT = invitation tournament; TC_2nd_ = second training camp; OT = official tournament.

**Table 4 ijerph-17-00775-t004:** The qualitative probabilities of daily session rating of perceived exertion and general wellness score in first training camp, invitation tournament, second training camp, and official tournament.

Parameters	TC_1st_	IT	TC_2nd_	OT	Qualitative Inferences for Effect Magnitude (Mean Difference; ± 90% CL)	
sRPE (a.u.)	553.15 ± 114.84	334.90 ± 44.21	906.32 ± 93.60	482.75 ± 81.22	Most likely large:	TC_1st_ vs. IT (209.81; 23.05); 100/0/0
						TC_2nd_ vs. OT (421.17; 1.9); 100/0/0
					Most likely small:	TC_1st_ vs. TC_2nd_ (-363.56; 8.30); 0/0/100
						IT vs. TC_2nd_ (-573.38; 1.20); 0/0/100
						IT vs. OT (-152.20; 12.50); 0/0/100
					Unclear:	TC_1st_ vs. OT (57.61; -); 50/0/50
Wellness (score)	16.74 ± 1.61	20.00 ± 3.18	16.63 ± 2.44	17.18 ± 0.98	Most likely large:	IT vs. TC_2nd_ (3.69; 1.6); 99.2/0.3/0.5
						IT vs. OT (2.81; 1.8); 97.6/1.0/1.3
					Most likely small	TC_1st_ vs. IT (-3.24; 0.5); 0/0/100
					Unclear:	TC_1st_ vs. TC_2nd_ (0.46; -); 50/0/50
						TC_1st_ vs. OT (-0.43; -); 50/0/50
						TC_2nd_ vs. OT (-0.89; -); 50/0/50

sRPE = session-rating of perceived exertion; SD = standard deviation; ES = effect size; CL = confident limit; TC_1st_ = first training camp; IT = invitation tournament; TC_2nd_ = second training camp; OT = official tournament.

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
