# Peer review of "Ultra-Short-Term and Short-Term Heart Rate Variability Recording during Training Camps and an International Tournament in U-20 National Futsal Players"

_ijerph, 2020, doi:10.3390/ijerph17030775_

Round 1

Reviewer 1 Report

The paper is well written, and their work is well presented. It should be published after minor revisions shown below.

Minor points.

Table 1 on a single sheet Modify table 2 graph the first 3 columns with error bar Modify table 3 graph the first 3 columns with error bar Modify figure 1 is not visible The Conclusions section by presenting the study limitations and suggestions for future research and by clearly stating the major implications of your study. The Conclusions section does not address any limitations of the study. Please consider listing and discussing the study limitations and how these limitations may be addressed in future studies. Future directions for additional research are not identified. Please consider whether future research and/or implementation recommendations should be provided.

Author Response

REVIEWER 1

The paper is well written, and their work is well presented. It should be published after minor revisions shown below.

Response: Thank you very much for your review and the comments. We have revised the manuscript according to the reviewers’ comments. Thanks again to review the manuscript and your comments to improve the quality of manuscript are appreciated.

Minor points.

Table 1 on a single sheet Modify table 2 graph the first 3 columns with error bar Modify table 3 graph the first 3 columns with error bar Modify figure 1 is not visible

Response:” We revised the tables according to the reviewer’s comment. In addition, we created a new figure in Figure 1 for LnRSSMDmean and LnRSSMDcv values in P5.

The Conclusions section by presenting the study limitations and suggestions for future research and by clearly stating the major implications of your study. The Conclusions section does not address any limitations of the study. Please consider listing and discussing the study limitations and how these limitations may be addressed in future studies. Future directions for additional research are not identified. Please consider whether future research and/or implementation recommendations should be provided.

Response: Thank you for bringing this up. The limitations of the study and future direction have been addressed in Line 309-322, P10.

Reviewer 2 Report

The aims of the present study was to examine ultra-short-term and short-term heart rate variability (HRV) in fourteen male young futsal players during pre-tournament training camps and an official tournament. The authors found that ultra-short-term record of LnRMSSDmean less than 1-min and short-term record of LnRMSSDcv at least 2-min during the training camps could be recommended in the futsal players. 

The manuscript presents the interesting data. However, the sample size was too small, which possibly affect the results (e.g. the correlations between training load, HRV, and wellness). Of LnRSSMDmean, I’d like to know why the variation of ES was pretty big. In addition, it is not easy to understand the coefficient of variation (CV) values of LnRSSMD (Table 3) were mostly over than 15% during the four study periods. Finally, the authors may consider using receiver operator characteristics (ROC) curve to validate the discriminant ability of HRV measures.

Minor comments

1. In Abstract, the subjects’ information needs to be added. 2. Regarding statistical analyses, please present how to calculate CV and how to interpret its reliability. 3. A Figure of Bland Altman plots seems necessary for data presentation. 4. Figure 1 is blurred.

Author Response

REVIEWER 2

The aims of the present study was to examine ultra-short-term and short-term heart rate variability (HRV) in fourteen male young futsal players during pre-tournament training camps and an official tournament. The authors found that ultra-short-term record of LnRMSSDmean less than 1-min and short-term record of LnRMSSDcv at least 2-min during the training camps could be recommended in the futsal players. 

The manuscript presents the interesting data. However, the sample size was too small, which possibly affect the results (e.g. the correlations between training load, HRV, and wellness). Of LnRSSMDmean, I’d like to know why the variation of ES was pretty big. In addition, it is not easy to understand the coefficient of variation (CV) values of LnRSSMD (Table 3) were mostly over than 15% during the four study periods. Finally, the authors may consider using receiver operator characteristics (ROC) curve to validate the discriminant ability of HRV measures.

Response: Thanks for your helpful feedback and for improving the quality of our manuscript.

The number of participants in this study was limited to 14 players due to final registered players of the official tournament (FIFA competition regulations) as we stated in the limitation of the study. We did have 21 players in-and-out across the training camps but only the best 14 were selected by the head coach for the tournament. As the purpose of this study investigated the ultra-short-term HRV comparison from preparation to the official tournament, we have to use the final 14 players for comparison.

For large effect size of LnRSSMDmean, the large and variant effects size was due to small standard deviation of LnRSSMDmean. We used inter-subject value to calculate the LnRSSMDmean, consequently small standard deviation. Based on previous studies (Flatt et al., 2017, J Sports Med Phys Fitness; Nakamura et al., in press, J Strength Cond Res), we recalculated the LnRSSMDmean by using intra-subject comparison. The statistical results showed trivial effect size in all-time segments (please refer to Table 2 in P5).

As mentioned above, we recalculated the LnRSSMDcv by using intra-subject comparison. The values of LnRSSMDcv were below 15% in the first study period and below 10% in the second, third, and forth study periods (please referring to Table 3 in P6).

Thanks for the suggestion to use the receiver operator characteristics curve in our study. The ROC method can provide sensitivity and specify of the measures in comparison. However, our research was within-subject design (one group of young male futsal players) in a longitudinal time period. We realized that it was not able to set a category to test the true and false cases for positive/negative results (Obuchowski, 2003, Radiology). We appreciated such valuable information that we will use this method to test the accuracy of the HRV measure in our future studies.

Minor comments

In Abstract, the subjects’ information needs to be added.

Response: We added the players’ information as “(age = 18.07 ± 0.73 yrs; height = 169.57 ± 8.40 cm; body weight = 64.51 ± 12.19 kg; body fat = 12.42 ± 3.18 %)”  in Line 24-25 in abstract.

Regarding statistical analyses, please present how to calculate CV and how to interpret its reliability.

Response: We added “The intra-subject coefficient of variation of LnRMSSD during each study period was used to calculate the LnRMSSDcv [37]”  in Line 163-164, P4-5 , to clarify how we measured the LnRMSSDcv in each period.

A Figure of Bland Altman plots seems necessary for data presentation.

Response:”. Dear reviewer, thank you. We do believe that the creation of bland altman plots for all will increase a lot the number of figures (20) and such is not necessary considering the main corrections that we made and the purpose of the study.

Figure 1 is blurred.

Response: We enlarged graphs and the size of scales and numbers in Figure 1.

Reviewer 3 Report

This manuscript presents heart rate variability data from 14 young adult male futsal players. The primary interest was in comparing abbreviated and non-abbreviated HRV sampling. The secondary interest was in assessing training load on HRV.

My primary concern with this manuscript is the small sample size and inclusion of males only. There is no sex-based hypothesis and thus no reason to exclude females. Presumably there is a U-20 female futsal team as well. In the 21st century there is no appropriate reason to include males only unless there is a sex-based hypothesis. The value of reporting differences (or non-differences) in 14 males to the larger literature is questionable at best.

Second, the authors hypothesize the null, i.e. that there will be "no difference" in the two sampling methods. There are certain ways to "test" the null, parameter estimates for example, but I don't believe the authors used those methods. (Please forgive me if the statistical method used does indeed allow for this. My default is to question any hypothesized non-difference.)

At the very least the first concern needs to be addressed or rectified before further consideration of this manuscript.

Author Response

REVIEWER 3

This manuscript presents heart rate variability data from 14 young adult male futsal players. The primary interest was in comparing abbreviated and non-abbreviated HRV sampling. The secondary interest was in assessing training load on HRV.

My primary concern with this manuscript is the small sample size and inclusion of males only. There is no sex-based hypothesis and thus no reason to exclude females. Presumably there is a U-20 female futsal team as well. In the 21st century there is no appropriate reason to include males only unless there is a sex-based hypothesis. The value of reporting differences (or non-differences) in 14 males to the larger literature is questionable at best.

Response: The purpose of this study was to investigate the degree of validity and agreement of ultra-short-term and short-term HRV records during 3 pre-tournament training camps and an official tournament in the U-20 national futsal players. The small sample size due to the limitation of the number of players registered in the final list of tournaments. This limitation has been responded to reviewer 2 and has been stated in P9. The reason we did not include female players because young female adults tournament at the national level has never existed in Asia. Therefore, it is impossible for us to collect such information in our region.

Second, the authors hypothesize the null, i.e. that there will be "no difference" in the two sampling methods. There are certain ways to "test" the null, parameter estimates for example, but I don't believe the authors used those methods. (Please forgive me if the statistical method used does indeed allow for this. My default is to question any hypothesized non-difference.)

At the very least the first concern needs to be addressed or rectified before further consideration of this manuscript.

Response: We revised the sentence according to reviewer’ comment: “Based on previous studies, we hypothesized that validity and agreement of ultra-short-term and short-term LnRMSSDs decrease during the study periods with high TL and low wellness scores”.

Round 2

Reviewer 2 Report

Thank you for a revision of the manuscript. But I don't think the authors answered my questions sufficiently. 

Author Response

Dear Reviewer, we have tried to improve the manuscript in all the points that we considered pertinent

Reviewer 3 Report

NA

Author Response

Dear reviewer

We have tried to improve the manuscript